# Effect of Combining Impact-Aerobic and Strength Exercise, and Dietary Habits on Body Composition in Breast Cancer Survivors Treated with Aromatase Inhibitors

**DOI:** 10.3390/ijerph20064872

**Published:** 2023-03-10

**Authors:** Marisol Garcia-Unciti, Natalia Palacios Samper, Sofía Méndez-Sandoval, Fernando Idoate, Javier Ibáñez-Santos

**Affiliations:** 1Department of Nutrition, Food Sciences and Physiology, Faculty of Pharmacy and Nutrition, University of Navarra, Campus Universitario, 31008 Pamplona, Spain; 2Center for Nutrition Research, University of Navarra, c/Irunlarrea 1, 31008 Pamplona, Spain; 3Navarra Institute for Health Research (IdiSNA), 31008 Pamplona, Spain; 4Centro de Estudios, Investigación y Medicina del Deporte (CEIMD), Gobierno de Navarra, 31005 Pamplona, Spain; 5Department of Gerontology and Public Health, Faculty of Health Science, University of Jyväskylä, Seminaarinkatu 15, Jyväskylän Yliopisto, 40014 Jyväskylä, Finland; 6Department of Physiology, Faculty of Health Science, Public University of Navarre, Av. Cataluña, s/n, 31006 Pamplona, Spain; 7Servicio de Radiología de la Mutua Navarra, 31012 Pamplona, Spain; fidoate@gmail.com

**Keywords:** combined exercise, impact-aerobic exercise, strength exercise, dietary habits, body composition, aromatase inhibitors, breast cancer survivors

## Abstract

This study examines both the effect of a twice-weekly combined exercise—1 h session of strength and 1 h session of impact-aerobic—on body composition and dietary habits after one year of treatment with aromatase inhibitors (AI) in breast cancer survivors. Overall, forty-three postmenopausal women with a BMI ≤ 35 kg/m^2^, breast cancer survivors treated with AI, were randomized into two groups: a control group (CG) (*n* = 22) and a training group (IG) (*n* = 21). Body composition, i.e., abdominal, visceral, and subcutaneous adipose tissue) was measured by magnetic resonance. In addition, some questionnaires were used to gather dietary data and to measure adherence to the Mediterranean diet. After one year, women in the IG showed a significant improvement in body composition, indicated by decreases in subcutaneous and visceral adipose tissue, and total fat tissue. Furthermore, the dietary habits were compatible with moderate adherence to the Mediterranean diet pattern and a low dietary intake of Ca, Zn, Folic Ac, and vitamins D, A, and E. A twice-weekly training program combining impact aerobic exercise and resistance exercise may be effective in improving the body composition for postmenopausal women who have breast cancer treated with AI, and the results suggest the need for nutritional counselling for this population.

## 1. Introduction

Breast cancer (BC) is defined as a malignant tumor that affects different breast cells [1]. It is considered a hormone-dependent disease characterized by molecular mechanisms involving activation of human epidermal growth factor receptor 2 (HER2), hormone receptors (estrogen receptor and progesterone receptor) and/or BRCA mutations [2]. Most BCs (70–80%) express a significant amount of estrogen receptors (ER) and/or progesterone receptors (PR), which are considered biomarkers of a favorable prognosis [3]. 

The disease can be classified according to the stage of the tumor and its localization, as well as according to the molecular subtype, determined by the analysis of the gene expression of HER2, and by quantitative hormone receptor (HR) [1]. 

Breast cancer is the most common tumor and is the first cause of death related to cancer among women [4]. According to estimates from Global Cancer Statistics, female breast cancer is the leading cause of global cancer incidence in 2020, representing 11.7% of all cancer cases, and is the fifth leading cause of cancer mortality worldwide. Among women, breast cancer accounts for 1 in 4 cancer cases and for 1 in 6 cancer deaths, ranking first for incidence and mortality in the majority of countries. This increase has been related to a higher prevalence of reproductive and hormonal risk factors and lifestyle risk factors, such as excess body weight and physical inactivity, as well as increased detection through mammographic screening [4]. In addition, the increasing incidence is linked to estrogen receptor-positive cancer, due to the stronger and more consistent association of excess body weight with estrogen receptor-positive cancer and the impact of mammographic screening, which preferentially detects slow-growing estrogen receptor-positive cancers [5,6]. 

The treatment depends on the woman’s hormonal status, kind, and stage of tumor, resulting in surgery therapy, mastectomy, radiation, chemotherapy, or if carcinoma is estrogen receptor–positive, patients may also receive endocrine therapy [7,8]. 

Early invasive stages (I, IIa, IIb) and locally advanced stages (IIIa, IIIb, IIIc) have three treatment phases: preoperative phase, surgery, and postoperative phase. The preoperative phase uses systemic endocrine or immunotherapies when tumors express estrogen, progesterone, or ERBB2 receptors. Preoperative chemotherapy may also be used and is the only option when tumors have none of those three receptors, and the postoperative phase includes radiation, immunotherapy, chemotherapy and/or endocrine therapy. When endocrine therapy is needed, only tamoxifen should be used in premenopausal women [9], while aromatase inhibitors for years is the current treatment recommended in postmenopausal women 35 years or older who are at increased risk for breast cancer and low risk for adverse medication effects [10].

Treatment with AI reduces cancer recurrence and improves overall survival in women in the early stages of the disease. However, this treatment is associated with musculoskeletal symptoms such as arthralgias, myalgias, osteoporosis-related bone fractures and unfavorable changes in body composition—a reduction of bone mineral density and muscle mass, and an increase in fat tissue and body weight [7]. Both have been related to poorer survival and an increase in cardiovascular events [11,12,13,14]. 

Given the efficacy of AIs and the large proportion of women diagnosed with breast cancer using this treatment, it is important to have available interventions to improve those AI side effects as well as the quality of life and all-cause mortality. Therefore, all women should be advised to adopt a healthier lifestyle that promotes overall health. In this context, although the evidence regarding the benefits of non-pharmacological measures specific to breast cancer survivors is limited. Recently, the American Cancer Society [15], and other authors [16,17,18,19] have published recommendations about healthy lifestyles that can help to prevent possible recurrences or complications. These include increasing physical activity, following a healthy diet pattern, and maintaining a healthy weight. In addition, there is some evidence about the potential negative side effects from AIs being diminished through the implementation of regular physical activity [20] and weight-bearing exercise, which includes impact and resistance training, following a healthy diet with adequate daily calcium and vitamin D intake [21,22,23,24]. We hypothesized that a low-frequency and twice-weekly training program combining one session of impact aerobic exercise and one session of resistance exercise might improve the body composition—body weight, body fat tissue and muscle mass—of postmenopausal women breast cancer survivors after one year of treatment with AI. 

Therefore, taking into account the need for more information, both on the most suitable exercise for patients treated with Ais and on their nutritional status and dietary habits, the aim of this study was two-fold. First, to analyze the effect of a low-frequency and twice-weekly training program combining one session of impact aerobic exercise and one session of resistance exercise on body composition—body weight, body fat tissue and muscle mass—of postmenopausal women with breast cancer after one year of treatment with AI; and second, to examine the lifestyle habits—nutritional and physical activity habits—within this population after one year of treatment with AI. 

## 2. Materials and Methods

### 2.1. Participants

This was a single-blind study that was undertaken for 5 years, from January 2012 to September 2017. A group of 43 postmenopausal women, aged 55–70 years, with a BMI of 19–35 kg/m², who had surgery for hormone-dependent breast cancer and treatment with chemotherapy and/or radiotherapy, and were about to begin treatment with AI, were included voluntarily in the study. The participants were referred by the Service of Oncology of a hospital in Spain. Performing physical exercise regularly and the presence of osteoporosis, diabetes mellitus, hypothyroidism, liver or kidney dysfunction, high blood pressure, heart disease, asthma, chronic obstructive pulmonary disease (COPD), joint disorders, alcoholism, or any other drug addiction, were considered as exclusion criteria, resulting in a sample of 43 women. The randomization procedure used prevented investigators from influencing group allocation. All researcher staff remained blinded until the end of the study, except for the sports trainers. The volunteers only were informed about their group assignment after the randomization process was completed. The volunteers were randomized using numbered and sealed envelopes that contained a paper with an intervention or control group. One-to-one, and sequentially, participants chose freely the envelope they wanted, resulting in being assigned to a Control Group (CG, 22 patients) or Intervention Group (IG, 21 patients), in which, each volunteer was followed for one year. Participant flow through the study is presented in Figure 1. 

### 2.2. Procedure

The CG and IG followed the usual medical supervision by the Service of Oncology and similar pharmacological treatment with AI and calcium (1000 mg/d) and vitamin D (20 µg/d) supplementation. During the time of the study, the IG followed a supervised training with two sessions per week: one session of odd-impact training (multidirectional impact aerobic exercise) and one of strength training. Both groups maintained their regular lifestyle and filled out a diary in which the volunteers recorded their physical activity, the appearance of symptoms, and the medication taken for the year of study. 

Further, anthropometric measures and tests of lifestyle, diet and nutritional status were carried out at the beginning of the study, before starting treatment with AI, and also at 6 and 12 months of the treatment.

### 2.3. Training Program

The training was divided into five training cycles in which the load was progressively increased: one cycle of 3 weeks of familiarization and four cycles of 9–14 weeks of duration. Each training session lasted between 60 and 90 min. There were two weeks of a break during the study, at weeks 31–32. The odd-impact training consisted of an aerobic choreography with multidirectional jumps. To increase the workload, the jump distance was increased from one-line jumps to 60 cm side squares jumps. The number of jumps per session varied from 96 during the familiarization period to 756 jumps at the end of the program. For strength training, different resistance machines were used (Technogym, Gambettola, Italy) for different exercises. The training workload began under 40% of 1-RM during the familiarization cycle and progressively was increased to 40–70% of 1-RM in the next four cycles. Each session consisted of 3 sets of 4–12 repetitions for 5 exercises. Two training routines were alternated. Each one had 3 common exercises—leg press, chest press and hip extension—and 4 exercises that were alternately performed according to the routines: Routine 1, which consisted of knee extension and pulldown; and Routine 2, which consisted of knee-flexion and shoulder press. Every 4–8 weeks, the 1-RM test was made to recalculate weights. All the sessions included warm-up, main part, and cool-down. During cool-down, stabilizing work of the abdominal-pelvic area and general stretching of worked-out muscles were made. All participants were required 90% to adhere to the training program. All the sessions were individually supervised by trained leaders. Besides transient musculoskeletal soreness, no major complications or injuries were reported.

### 2.4. Body Composition and Nutritional Status 

The study of body composition involved body weight, muscle mass and fat tissue through anthropometric measurement of body weight, height, waist and arm circumferences, and triceps skinfold (TSF). From these data, we calculated the Body Mass Index (BMI, kg/m^2^), total body fat (% BF) according to the CUN-BAE formula [25], the arm muscle circumference (AMC), and the arm muscle area (AMA) [26]. The measurements of these parameters were made by the same dietitian, following the protocol of the International Standards for Anthropometric Measurements of ISAK [27]. Patients were weighed in light clothing without shoes on SECA 714 weighing scale with a graduation of 0.1 kg. Patient height was determined with a height rod stadiometer (SECA 220) using a graduation of 1.0 mm. All measurements were obtained in duplicate. The measurements of waist, hip, and arm circumferences were made in duplicate or triplicate using a Cescorf flexible steel tape measure with a graduation of 1.0 mm. The triceps skinfold thickness was measured using a Harpenden skinfold calliper (0.2 mm). The participants were classified according to their BMI as described by World Health Organization [28]. Moreover, abdominal visceral adipose tissue (VAT) (cm^2^) and abdominal subcutaneous adipose tissue (SAT) (cm^2^) were measured at L3 discal level. The muscle volume was measured in the thigh. These three measurements were made by magnetic resonance (Magnetom Impact Expert; Siemens) by the same experienced operator using a body coil [29]. Intra-observer reliability for calculation of total VAT and SAT volumes was 0.99 with a coefficient of variation of 5–8%. The Patient-Generated Subjective Global Assessment (PG-SGA) was applied during a face-to-face interview. According to the PG-SGA score, patients were classified from 0 to 8 with adequate nutritional status and ≥9 with undernutrition status [30]. 

### 2.5. Level of Physical Activity

The total level of physical activity was estimated in terms of metabolic equivalents (METs). We collected information about physical activity through a 17-item leisure time physical activity questionnaire, which provided valid and reproducible self-reported data [31,32,33]. METs were calculated according to the activity and the time spent on it per week, obtaining METs/h/week. In the IG, we added 3 additional METs. These correspond to the estimate of the energy expenditure of the strength training session of moderate intensity, carried out by the volunteers in this group [34,35]. 

### 2.6. Dietary Habits 

The study of dietary habits was conducted by a trained nutritionist. It included the study of nutrient intake through non-consecutive 3-day food records [36]. To quantify food and beverage consumption at home, the volunteers used a scale (Leifheit), while to estimate the intake outside their homes, they used a photo album [37]. Additionally, the intake of vitamins and minerals supplements was recorded. The food weights were converted to nutrient intake estimates per day by the EasyDiet^®^ program [38], using the Spanish food composition database. The study of the dietary pattern was assessed using a food frequency questionnaire validated in Spain [31,32,33] as well as the Mediterranean Diet Score (MDS) [39], which assesses adherence to the Mediterranean dietary pattern (MedDiet). For the present study, the score obtained in MDS was classified into three categories: with 0–2 points indicating low adherence, 3–6 points indicating moderate adherence, and 7–9 points indicating high adherence to the MedDiet [40]. Furthermore, diet quality was assessed according to the recommended food servings and the nutritional goals authorized by the Spanish Society for Community Nutrition (SENC) [41,42]. To carry out the analysis of the adequacy of intake to the recommendations, it was established that the volunteers met the recommendation if they reached the minimum indicated in the guidelines of the SENC. For products classified as “Occasional and moderate consumption”, the recommended weekly serving was established as one.

### 2.7. Statistical Analyses

STATA statistical software package version 12.1 was used for the statistical analysis (StataCorp. (2011). Stata Statistical Software: Release 12 College Station, TX: StataCorp LP). On the one hand, two-sided analysis and a level of 5% using the *t*-test for independent samples and *t*-test for matched data for the parametric variables were used. On the other hand, the U of Mann-Whitney sign test or Wilcoxon rank test were used for non-parametric variables. For qualitative variables, frequency analysis was used. Data are presented as mean (SD) for quantitative variables and as a percentage of sample *n* (%) for qualitative variables.

## 3. Results

At baseline, no significant differences were found between volunteers in the CG and IG. The only exception was the level of physical activity, which was significantly higher in the CG (Table 1). It should be noted that none of the participants were at risk of malnutrition; on the contrary, some level of overweight was observed (BMI > 24.9 kg/m²) in 63% of the women (Appendix A). Biochemical data are shown in Appendix A.

### 3.1. Body Composition and Nutritional Status

At the end of the study, the IG showed a decrease in weight, subcutaneous fat measured by the triceps skinfold (TSF), and waist circumference, associated with a decrease in abdominal fat, both subcutaneous (SAT) and visceral (VAT). Furthermore, the decrease of two points in PG-SGA is interpreted as an improvement in the nutritional status.

In contrast, among the volunteers in the CG, there were no significant changes in the nutritional status or body composition, although a tendency towards an increase in total fat was observed (*p* = 0.07). When both groups were compared, significant differences were observed at the end of the study in weight, BMI, waist circumference, %Total BF, TSF, and VAT, in the expected direction of the intervention; more details are shown in Table 2, Figure 2 and Figure 3. 

### 3.2. Physical Activity Level

At baseline, the participants in the CG presented a higher level of physical activity than those in the IG. However, after one year of follow-up, we observed a clear tendency to a decrease in the total physical activity level of the CG (−11.8 METs/h/week; *p* = 0.07). Whereas, in the IG, the level of total physical activity was maintained, although the exercise program was equivalent to 10–15 METs/h/week. No differences between the two groups were observed in the total physical activity level at the end of the study (Table 2).

### 3.3. Dietary Habits

According to SENC guidelines [41,42], both groups of volunteers had lower cereal and alcohol consumption, while the intake of fish was higher than the recommendations. For the IG, consumption of red meat, fast food, pastries, and sweetened soft drinks was also higher than the recommendations. The two groups showed a trend towards low fruit consumption (Table 3). Nevertheless, these results are in line with those obtained from the Mediterranean Diet Score (MDS). After 12 months of intervention, we observed only around 19% of the total volunteers presenting a high adherence to the MedDiet pattern and 77% of them having moderate adherence to the pattern (Appendix A).

After one year of follow-up, no changes in caloric intake and macronutrient intake were observed in any of the two groups of the study (Appendix A). We only saw an increase in the intake of some micronutrients—total vitamin D in the CG (mean difference of 5.9 μg/d ± 13.1 μg/d; *p* < 0.05) and the iron (mean difference of 1.9 mg/d ± 3.8 mg/d; *p* < 0.05) and niacin (mean difference of 3.2 mg ± 5.0 mg; *p* < 0.05) intake in the IG (Appendix A). 

## 4. Discussion

### 4.1. Body Composition and Nutritional Status

The main finding of this study was that a twice-weekly training program combining one session of impact-aerobic exercise and one of resistance training in women treated with breast cancer with AI led to significant weight and fat weight loss, total fat loss, and decreased waist circumference, associated with a significant SAT and VAT loss. In contrast, the CG did not show any variation in body composition. When overweight, VAT and SAT volumes are positively related to cardiovascular risk, and it is known that cardiovascular disease is an important cause of death in patients with breast cancer [43,44]. Furthermore, although there is limited evidence that intentional weight loss after diagnosis may be beneficial for breast cancer-specific mortality, some studies suggest that weight loss over this time frame may be suggestive of a lower relative risk of breast cancer recurrence in estrogen receptor (ER)-positive breast cancer survivors [45]. Furthermore, in this context, it has been shown that therapy with AI in sedentary breast cancer patients is related to changes in fat distribution, with a relatively great VAT/SAT ratio, regardless of whether they gain or lose weight after therapy [46], and this pattern of fat distribution is associated with breast cancer recurrence [47]. Dimauro et al. [1] studied the effect of different physical exercise protocols as secondary and tertiary prevention among ER+ breast cancer survivors before and during pharmacological treatments, including aromatase inhibitors. The protocols included two or more training days per week, two or more days of resistance training per week, the inclusion of high-intensity (HIIT) training, or the combination of diet with the training exercise, with favorable results in body composition and decrease in the side effects of aromatase inhibitors in old BC patients. However, unlike our study, none of them included only one session of impact-aerobic exercise and one of resistance training without a concomitant hypocaloric diet. Thus, the improvements in body composition observed in this study show that this novel protocol, performed twice a week, could mitigate the risk of cardiovascular disease and breast cancer recurrence risk in postmenopausal women treated for breast cancer with AI [15].

However, our training program did not provide enough stimulus to achieve significant improvements in muscle mass. A study by Thomas et al. [48], which included three sessions of exercise per week, two of which combined strength and aerobic exercise and one session of 150 min of moderate-intensity exercise at home over 12 months, showed losses in fat mass as well as an increase in lean mass in cancer survivors treated with AI. The reason for this difference in our study may be related to the number of strength training sessions per week. Our training program consisted of only two days per week of scheduled activity, and only one of those days was dedicated to a strength workout. Unlike our study, the one carried out by Santos et al. [49] investigated the effects of resistance training, once a week for 8 weeks, on changes in body composition and muscular strength in breast cancer survivors, and although no changes in body composition were detected, they observed improved muscular strength [49].

These results suggest that although exercise programs equivalent to 10–15 METs/h/week may result in significant fat loss, it would be necessary to perform at least two strength-training sessions per week to achieve significant improvements in muscle mass in postmenopausal breast cancer survivors treated with AI.

### 4.2. Lifestyle: Physical Activity Level and Dietary Habits

The 2018 American Physical Activity Guide showed that regular recreational physical activity is associated with a decrease in the risk of breast cancer death and mortality from any other cause [50]. At the beginning of our trial, the group of volunteers randomly assigned to the CG presented a level of physical activity 1.6 times higher than the IG, but the physical activity in both groups was considered moderate [51,52]. During the year of the study, the CG continued with their usual activities while the IG attended two weekly training sessions. However, at the end of the study, it was observed that the tendency in the CG was to decrease the level of physical activity. Meanwhile, for the IG, thanks to the programmed exercise, the physical activity was maintained, bringing both groups in line with each other. The behavior observed in the CG is aligned with that described in the Women Health Initiative study, in which results showed that the level of physical activity decreased over time, maintaining significantly low levels even 10 years after a breast cancer diagnosis. The tumor stage at diagnosis, the type of treatment received, the BMI, and the presence of other comorbidities were identified as possible causes of the decrease in physical activity [53]. In our study, both groups of volunteers reported an increase in arthralgias and myalgias associated with AI treatment, so this may be another factor to consider when explaining the decrease in the level of physical activity in the CG. To note, our training program resulted in a metabolic output within 10–15 METs/h/week, depending on the phase and intensity of training, which seems to be effective in counteracting the reduction in total physical activity level that spontaneously occurs in some women with cancer treated with AI. Some authors have found that the practice of physical activities equivalent to 10 METs/h/week or more, i.e., 2.5 h/week at moderate intensity, is associated with a 24–27% reduction in all-cause mortality and a 25% reduction in breast cancer-specific mortality [54,55], improving overall survival after cancer diagnosis [56]. Recently, two systematic reviews and meta-analyses [57,58] compared women who had the highest levels of recreational or total physical activity with those who had the lowest levels and found reduced risks of 42% all-cause and between 37% and 42% breast cancer-specific mortality in the most active versus least active categories. “The protective effects of physical activity on breast cancer-specific mortality may include reduced exposure to estrogen and androgen, the effects of insulin and insulin-related factors, and reduced inflammation. Physical activity may affect these pathways directly or indirectly by its effects on reducing body weight. The lower risk for all-cause mortality may be linked to other benefits of physical activity through reduced cardiovascular risk (i.e., improved exercise capacity) and reduced risk for other comorbidities” [59]. Nevertheless, currently, there is insufficient information to support specific recommendations related to the domain, optimal dose and timing of activity that could contribute to the dose of 10–15 METs hours per week in postmenopausal women with breast cancer [15,57,60]. The evidence highlights the importance of breast cancer survivors engaging in any amount of physical activity they can, increasing their activity level when possible, and especially not decreasing physical activity after their diagnosis and treatment [15]. In this context, the training exercise protocol that we propose fits with the amount of physical activity recommended by these groups and leads us to consider that our program of intervention could be an essential part of the treatment.

Dietary habits can also play a relevant role in body composition and the risk of cardiometabolic diseases [61,62,63,64,65,66,67,68], and evidence shows that high adherence to MedDiet reduces the risk of chronic disease (including cardiovascular disease), cancer mortality, and appears to have a preventive effect on the incidence of breast cancer [67,69,70,71,72,73]. An increase of only two points in the MDS has been associated with a 25% reduction in total mortality [39]. In cancer survivors, there is little evidence indicating that dietary behaviors influence outcomes regarding recurrence and mortality. However, even though the evidence is limited, studies suggest that a high-quality diet such as MedDiet is beneficial for these patients [70,71] due to the presence of bioactive compounds that could act by different mechanisms, such as increasing antioxidant capacity, controlling body weight, and improving the glucose profile [65,68,70,73,74,75,76,77,78]. For this reason, knowing the eating habits of a population can provide more effective nutritional advice.

In our study, although in both groups the consumption of most food groups was in line with the SENC recommendations about a healthy diet, the percentage of volunteers with high adherence to MedDiet decreased from 20.8% to 18.6% during the year of the study, and this was reflected in the frequency of consumption by food groups. In general, the consumption of red and processed meats, fast food, pastries, and soft drinks was higher than the recommendations, and this intake has been linked to an adverse effect on body weight control, cardiovascular risk, and cancer risk, due to their higher caloric intake and their content of saturated fat and cholesterol [62,63,64,65,66,68]. Moreover, high consumption of saturated fat, from red and processed meat, has been associated with an increased risk of hormone receptor-positive breast cancer and breast cancer-specific mortality [62,76,79]. Although these results are aligned with those observed in a recent study carried out in the north of Spain, as well as with the general pattern of consumption among the Spanish population [41,80], they differ from those obtained by other researchers, who showed dietary changes among breast cancer survivors in the first year after treatment with a lower intake of red meat consumption along with lower energy [81,82,83,84]. One aspect that appears to influence the acquisition of healthier lifestyle habits in patients may be professional advice. Recommendations by health professionals are associated with higher rates of behavior change among cancer survivors [85,86]. In this study, no nutritional intervention was carried out, which could justify the minimal changes observed in the dietary intake of our sample.

Concerning the intake of macro and micronutrients, it was barely modified in the whole sample of volunteers during the year of the study, but it is also worth noting that the dietary intake of Calcium, Zinc, Folic acid and vitamins D, A and E did not cover the recommendations for this age group in the Spanish population [87]. The data obtained were similar to those reported by the ANIBES study [80,88] in Spanish women with a similar age range. However, the requirements of calcium and vitamin D were ensured by the prescription of pharmacological supplements since the beginning of the intervention. Previous studies have shown that low intakes of Calcium and vitamin D may contribute to low serum vitamin D levels, which are inversely related to breast cancer [89,90]. On the contrary, higher levels of these elements are related to increased survival, as well as an improvement in bone health [91]. This is especially relevant in patients treated with AI, where significant losses in bone mineral density have been reported. Although no evidence directly relates vitamins A and E intake with susceptibility or prevention of breast cancer, the antioxidant effect of both vitamins could act positively on cell damage and cell activity, aspects associated with the development and recurrence of cancer [91]. Concerning Folic Acid, the results of the EPIC cohort showed that folate status was not clearly associated with breast cancer risk [92,93].

In summary, according to the literature, lifestyle interventions that include an increased level of physical activity and nutritional counselling appear to be useful to improve therapeutic efficacy, limiting drug-induced side effects, including those related to AI [94], and improving overall survival after cancer diagnosis [56]. This leads us to consider, that these interventions should be an essential part of the therapeutic approach [76]. This suggests the need for professional counselling, so it would be important for health professionals to provide timely education to allow patients to perceive the benefits of healthier lifestyles throughout the treatment period [86,95].

### 4.3. Limitations of the Study

The main study limitation was the magnitude of the trial, which limited the ability to draw strong conclusions. Other limitations were the differences in physical activity between the groups before enrollment, but it was a description of the real physical activity of volunteers, not related to the practice of any programmed physical activity; and the use of food records to study dietary habits. However, dietary records are considered a reference method in validation studies.

## 5. Conclusions

This study shows that a twice-weekly training program combining one session of impact-aerobic exercise and one session of resistance exercise improved the body composition in postmenopausal women treated for breast cancer with AIs. After one year, without a concomitant weight loss diet or dietary advice, women in the exercise group showed a decrease in total fat, SAT, VAT, and waist circumference. This low-frequency combined exercise program may be more acceptable, effective, and practical in optimizing aspects of physical fitness than programs that involve a higher frequency. It is suggested that the choice of this low-frequency regimen will be made when the ability of the individual to withstand a vigorous program or the willingness to devote a high weekly training frequency is a constraint. In addition, the dietary habits of our volunteers were characterized by dietary habits compatible with moderate adherence to the Mediterranean diet pattern, but with a low dietary intake of Ca, Zn, Folic Ac, Vit D, A and E, which suggests the need for nutritional counselling. All these issues should be considered when planning future advice programs for breast cancer survivors.

## Figures and Tables

**Figure 1 ijerph-20-04872-f001:**
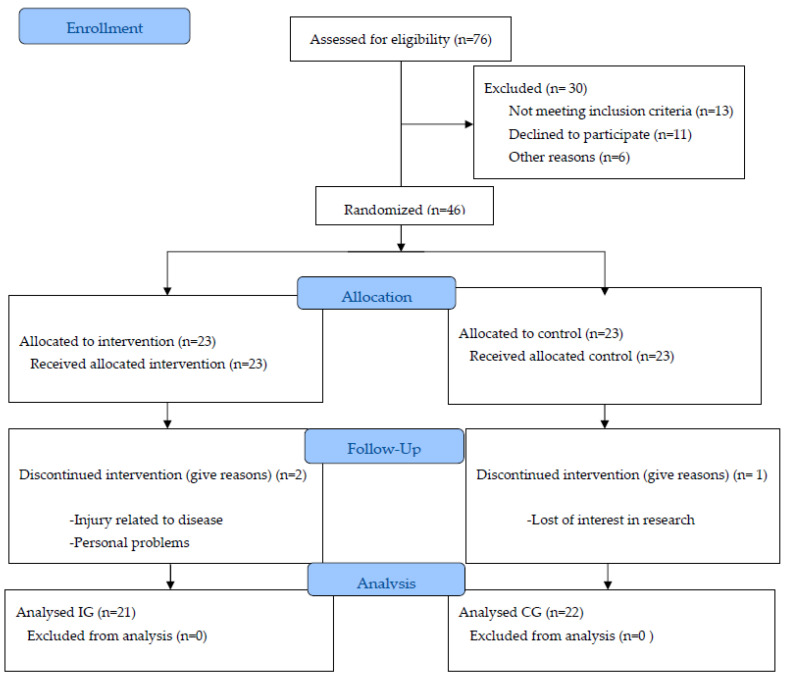
Flow chart. The flow of participants throughout the study protocol. Notes: IC—intervention group; CG—control group; n—number of participants.

**Figure 2 ijerph-20-04872-f002:**
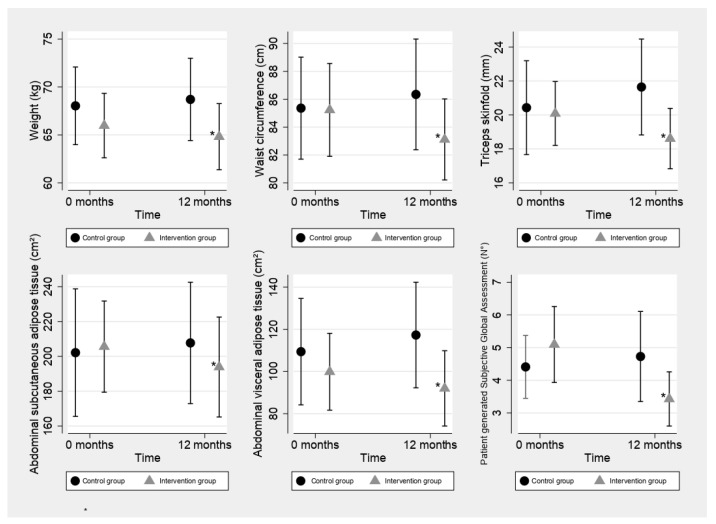
Changes in the body composition and nutritional status after twelve months of study. * Significant difference (*p* ≤ 0.05). Each box represents the changes observed between the Control Group (Control) and Intervention Group (Intervention) from baseline. Note, that the Intervention Group significantly improved all variables of body composition, as well as the nutritional status after the training program.

**Figure 3 ijerph-20-04872-f003:**
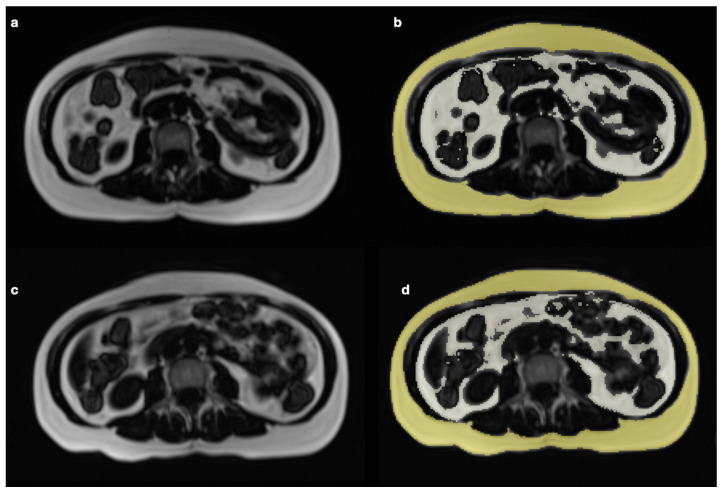
Pre-intervention (**a**,**b**) and post-intervention (**c**,**d**) magnetic resonance (MR) images were obtained at the L3 level. Adipose tissue appears white and non-adipose tissue appears dark on non-segmented images (**a**,**c**). Segmented images show visceral fat (VAT) in grey and subcutaneous fat (SAT) in yellow color. A decrease in subcutaneous fat and visceral fat is clearly observed.

**Table 1 ijerph-20-04872-t001:** Baseline characteristics and body composition of the Control Group (CG) and Intervention Group (IG) of the study.

	Control Group (CG)		Intervention Group (IG)		CG vs. IG
	Mean	SD	Mean	SD	*p*-Value
Age (years)	61.54	5.19	60.48	4.96	0.494
Tumor stage, *n* (%)	*n*	(%)	*n*	(%)	0.495
Stage I	14	63.6	17	81.0	
Stage II	7	31.8	3	14.3	
Stage III	1	4.5	1	4.8	
Treatment, *n* (%)	*n*	(%)	*n*	(%)	0.251
Surgery	3	13.6	0	0.0	
Radiotherapy	9	40.9	13	61.9	
Chemotherapy	3	13.6	1	4.8	
Radiotherapy and Chemotherapy	7	31.8	7	33.3	
Score PG-SGA (SD)	4	2.0	5	3.0	0.417
Nutritional status according to BMI, *n* (%)	*n*	(%)	*n*	(%)	1.000
Underweight	0	0.0	0	0.0	
Normal weight	8	36.4	8	38.1	
Overweight	9	40.9	9	42.9	
Obesity Class I	4	18.2	4	19	
Obesity Class II	1	4.5	0	0.0	
Obesity Class III	0	0.0	0	0.0	
Body composition	Mean	SD	Mean	SD	*p*-value
Weight (kg)	68.0	9.7	66.0	7.9	0.451
BMI (kg/m^2^)	26.3	4.3	26.2	3.1	0.921
% BF	39.7	4.9	39.5	3.8	0.921
Waist circumference (cm)	85.4	8.8	85.2	7.8	0.961
Waist hip ratio	0.84	0.06	0.83	0.07	0.855
Triceps skin fold (mm)	20.4	6.6	20.1	4.4	0.842
SAT (cm^2^)	206.1	81.9	205.6	56.7	0.982
VAT (cm^2^)	111.0	57.3	99.9	39.4	0.501
AMC (cm)	23.8	2.3	23.7	1.7	0.849
AMA (cm^2^)	38.9	8.0	38.27	6.4	0.792
MT (cm^2^)	45.0	7.1	41.2	5.6	0.088
Level of physical activity (METs/h/week)	42.5	25.6	26.2	20.2	0.027 *

Notes: Data are presented as means and standard deviation (SD) or number (*n*) and percentage of participants (%). * Significant difference between CG and IC, (*p* ≤ 0.05). The significant *p*-value is shown in bold. Abbreviation: SD—standard deviation; PG-SGA—patient-generated subjective global assessment; BMI—body mass index; BF—total body fat; SAT—abdominal subcutaneous adipose tissue; VAT—abdominal visceral adipose tissue; AMC—arm muscle circumference; AMA—arm muscle area; MT—thigh muscular tissue; METs—metabolic equivalent.

**Table 2 ijerph-20-04872-t002:** Changes in body composition, nutritional status, and physical activity in the Control Group (CG) and Intervention Group (IG) after one year of the study.

	Control Group (CG)			Intervention Group (IG)			CG vs. IG
	Mean Difference †	SD	*p*-Value	Mean Difference †	SD	*p*-Value	*p* Mean Difference Cg-IG ‡
Weight (Kg)	0.7	3.0	0.302	−1.2	2.4	0.042 *	0.033 *
BMI (kg/m^2^)	0.3	1.1	0.244	−0.4	1.0	0.057	0.031 *
% BF	0.4	1.3	0.067	−0.4	1.2	0.147	0.031 *
Waist circumference (cm)	1.0	3.3	0.172	−2.1	4.0	0.026 *	0.008 **
Waist Hip Ratio	0.0	0.0	0.449	−0.0	0.0	0.128	0.085
Triceps skinfold (mm)	1.2	3.8	0.286	−1.5	3.1	0.039 *	0.014 *
SAT (cm^2^)	0.9	35.8	0.912	−11.7	17.1	0.010 *	0.188
VAT (cm^2^)	7.8	26.5	0.227	−7.9	15.3	0.043 *	0.036 *
AMC (cm)	−0.2	1.2	0.431	0.1	1.0	0.704	0.395
AMA (cm^2^)	−0.9	4.9	0.406	0.3	3.7	0.731	0.386
MT (cm^2^)	−0.4	6.0	0.760	1.6	3.5	0.078	0.121
PG-SGA Score	0	3	1.000	−2	3	0.019 *	0.067
METs/h/week	−11.8	29.2	0.073	3.4	23.2	0.516	0.068

Notes: Data are presented as means of the difference from baseline, together with the standard deviation (SD). † Within-group, mean difference in parameter, between the beginning and end of the study. ‡ the comparison of differences in the parameters at the end of the study between CG and IG and the p-value. * Significant difference (*p* ≤ 0.05). ** Significant difference (*p* ≤ 0.001). The significant *p*-value is shown in bold. Abbreviation: SD—standard deviation; BMI—body mass index; BF—total body fat; SAT—abdominal subcutaneous adipose tissue; VAT—abdominal visceral adipose tissue; AMC—arm muscle circumference; AMA—arm muscle area; MT—thigh muscular tissue; PG-SGA—patient-generated subjective global assessment; METs—metabolic equivalent.

**Table 3 ijerph-20-04872-t003:** Changes from the baseline in the Control Group (CG) and Intervention Group (IG) in food consumption patterns and adequacy of food frequency to the recommendations proposed by the Spanish Society of Community Nutrition (SENC).

		Control Group (CG)									Intervention Group (IG)									
		Initial			Final			(a)			Initial			Final			(a)			(b)
Foods	SENC	Mean	SD	*p*-Value	Mean	SD	*p*-Value	Mean	SD	*p*	Mean	SD	*p*-Value	Mean	SD	*p*-Value	Mean	SD	*p*	*P* ♯
Cereals	4–6	2.4	1.6	**0.001 ****	3.4	3.3	**0.017 ***	1.0	3.1	0.664	2.4	2.1	**0.002 ***	2.6	1.8	**0.007 ***	0.3	2.5	0.115	0.771
Vegetables	≥2	2.2	0.9	0.211	2.4	1.2	0.094	0.2	1.0	0.832	2.2	0.8	0.346	2.7	1.6	0.664	0.5	1.6	0.503	0.568
Fruit fresh	≥3	2.7	1.8	0.053	2..8	2.2	0.053	0.1	1.7	1.000	2.2	1.1	**0.004 ***	2.4	1.5	0.069	0.1	1.4	0.664	0.961
Pulses	2–3	2.5	0.9	**0.015 ***	2.2	0.9	0.436	−0.4	1.0	0.085	2.0	1.1	0.263	2.0	1.0	0.078	−0.1	1.4	0.860	0.376
Nuts	3–7	2.0	2.2	0.078	3.1	4.2	0.064	1.0	4.3	0.077	1.8	2.8	**0.007 ***	2.3	2.4	0.144	0.5	3.3	0.332	0.854
Dairy	2–4	2.8	1.8	0.524	2.8	1.9	0.832	−0.1	2.0	0.908	2.7	1.7	0.189	2.5	1.2	0.087	−0.2	1.9	0.664	0.961
Fish	2–3	5.6	2.3	**0.001 ***	6.0	3.1	**0.001 ***	0.4	2.6	0.495	5.8	2.9	**˂0.001 ***	5.3	2.1	**˂0.001 ***	−0.5	2.5	0.399	0.281
Poultry	3–4	2.5	1.4	0.132	2.4	1.3	0.051	−0.1	1.5	0.791	2.6	1.8	0.272	2.7	1.7	0.420	0.1	1.1	0.585	0.595
Red meat	≤1	2.7	1.6	**0.001 ***	2.5	2.5	0.383	−0.2	2.1	**0.049 ***	2.0	1.3	**0.001 ***	2.1	1.6	**0.019 ***	0.1	1.5	1.000	0.208
Eggs	3–5	2.8	1.5	0.576	2.5	1.2	0.125	−0.3	1.1	1.000	2.5	1.0	0.063	3.5	3.5	1.000	1.1	3.7	0.625	0.252
Fast food, pastries, soft drinks	≤1	3.8	4.0	0.286	3.7	4.4	0.286	−0.1	4.1	0.824	4.4	4.4	**0.012 ***	5.0	4.4	**0.007 ***	0.5	5.4	0.629	0.618
Alcoholic drinks	1	0.3	0.4	**0.001 ***	0.4	0.5	**0.001 ***	0.1	0.3	0.267	0.4	0.5	**˂0.001 ***	0.5	0.8	**˂0.001 ***	0.1	0.4	0.581	0.871

Notes: Data are presented as the mean frequency of consumption by food groups: cereals—services/day; vegetables—services/day; fruit fresh—services/day; pulses—services/week; nuts—services/week; dairy—services/day; fish—services/week; poultry—services/week; red meat—services/week; eggs—services/week; fast food, pastries, soft drinks—services/week; alcoholic drinks—services/day. First, the table shows the recommendations proposed by the Spanish Society of Community Nutrition (SENC) on the frequency of consumption by food groups—SENC. Second, for each group—CG and IG—the intake of the volunteers expressed as the Mean and deviation standard (SD), and the values of the comparison (*p*-value) between the real intake and the SENC recommendations. Third, (a) represents within-group comparison of differences in food consumption, between the beginning and end of the study, for each food group. Finally, (b) represents the comparison of differences in food consumption at the end of the study between CG and IG and the *p*-value—*P ♯*. CG—Control Group, IG—Intervention Group; SD—deviation standard. * Significant difference (*p* ≤ 0.05). ** Significant difference (*p* ≤ 0.001). The significant *p*-value is shown in bold.

## Data Availability

The data presented in this study are available in Appendix A.

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
