# Peer review of "Effect of Combining Impact-Aerobic and Strength Exercise, and Dietary Habits on Body Composition in Breast Cancer Survivors Treated with Aromatase Inhibitors"

_ijerph, 2023, doi:10.3390/ijerph20064872_

Round 1

Reviewer 1 Report

I think the manuscript is interesting and deserving of publication. There are, however, some comments that I consider minor and I'd like to recommend the authors to take in consideration.

  1. The introduction is too small; a short paragraph that describes breast cancer and its symptoms and diagnosis is helpful with recent references.
  2. The author must explain the choice of aromatase inhibitors (AI), and postmenopausal women?
  3. Is the number of patients sufficient to carry out this study (43 patients)
  4. In retrospective studies, the samples must be varied from different regions, in this manuscript the choice of patients from a single region "Espagne", please explain.
  5. Examples of X-rays or ultrasounds of the patients are necessary to confirm the results obtained.
  6. There are some biochemical parameters especially lipid profile should be added in this study.
  7. All figures and tables should be consistent and professionally drawn. The fonts are different and some of the images are stretched. 
  8. Figure 2: please add statistical (*) in the legend.

Reviewer 2 Report

The manuscript titled “Effect of Combining Impact-Aerobic and Strength Exercise, and Dietary
Habits on Body Composition in Breast Cancer Survivors Treated with Aromatase Inhibitors” by  Marisol Garcia-Unciti et al., is the outcome of a trial which was undertaken to examine the effects of a twice-weekly combined exercise (strength exercise, 1h/session/week and one impact-aerobic exercise, 1h/session/week) performed for one year, on body composition and the dietary habits of postmenopausal women who were survivors of the breast cancer (n=43, 21 in the intervention group, and 22 in the control group) who were being treated with aromatase inhibitors. The researchers reported that after one year the women in the intervention group showed a significant improvement in body composition, and a decrease in subcutaneous and visceral adipose tissue, and total fat tissue. In addition, the nutritional status of the participants in the intervention group improved at the end of study. Please see the following as my questions or suggestions for the authors:

11. In the first line after the section title “Introduction”, please clarify whether breast cancer is the most common type of cancer worldwide. Also, I suggest you provide referenced data regarding the incidence and prevalence of this cancer worldwide.

22. In the first paragraph of the Materials and Methods, in the term “Chronic obstructive pulmonary disease”, the word “chronic” should not be started with a capital letter.

33. What was the reason for choosing 43 patients? Was a sample size calculation used? If not, was this a pilot study? If so, please specify that.

44. Under the title “participants”, please specify what method was used for randomisation?

55. In the text, please clarify that whether the patients (IG and CG) were aware of their group allocations?

66. Figure 1 has technical issues (words have been cut out from the boxes).

77. Please correct the spelling error in Figure 1 (i.e., losse)

88. In the text, please specify over which time period this study was undertaken.

Overall, I believe this is a good study and can be improved by addressing the above-mentioned comments.

Round 2

Reviewer 1 Report

Accept

Author Response

We appreciate the positive comments of the reviewer about our manuscript.

Reviewer 2 Report

I can see the manuscript has improved by some of the changes made to it. However, I would like to ask the corresponding author that please provide short and sharp answers to my questions rather than writing a whole paragraph for each question and asking me to find my answer in it. Actually, I read through the provided paragraphs, but I was unable to find answers to some of my questions, for example, what was the study period and what was the randomisation mothed for allocating participants to IG or CG? Please revisit all my questions and provide a specific answer for each question. Thank you.

Author Response

We apologize for not being more specific in our responses.  We will try to answer your questions more specifically. 
